DOI: 10.1038/s41467-017-02760-1　　**OPEN**

# BLM helicase suppresses recombination at G-quadruplex motifs in transcribed genes

Niek van Wietmarschen[1], Sarra Merzouk[1], Nancy Halsema[1], Diana C.J. Spierings[1], Victor Guryev[1] & Peter M. Lansdorp[1,2,3]

Bloom syndrome is a cancer predisposition disorder caused by mutations in the *BLM* helicase gene. Cells from persons with Bloom syndrome exhibit striking genomic instability characterized by excessive sister chromatid exchange events (SCEs). We applied single-cell DNA template strand sequencing (Strand-seq) to map the genomic locations of SCEs. Our results show that in the absence of BLM, SCEs in human and murine cells do not occur randomly throughout the genome but are strikingly enriched at coding regions, specifically at sites of guanine quadruplex (G4) motifs in transcribed genes. We propose that BLM protects against genome instability by suppressing recombination at sites of G4 structures, particularly in transcribed regions of the genome.

[1] European Research Institute for the Biology of Ageing, University of Groningen, University Medical Center Groningen, Antonius Deusinglaan 1, 9713 AV Groningen, The Netherlands. [2] Terry Fox Laboratory, British Columbia Cancer Agency, Vancouver, BC V5Z 1L3, Canada. [3] Department of Medical Genetics, University of British Columbia, Vancouver, BC V6T 1Z4, Canada. Correspondence and requests for materials should be addressed to P.M.L. (email: plansdor@bccrc.ca)

Bloom syndrome (BS) is a rare genetic disorder caused by mutations in the *BLM* gene, which encodes the BLM helicase[1]. Symptoms of the disease include short stature, immunodeficiency, UV sensitivity, reduced fertility, and a strong predisposition toward a wide range of cancers. Cells from BS patients display marked genome instability, characterized by a 10-fold increase in the rate of sister chromatid exchange events (SCEs) in cells from patients compared with healthy controls[2,3]. SCEs are a byproduct of double-strand breaks (DSBs) or collapsed replication forks that are repaired via homologous recombination (HR)[4,5]. Although SCEs are typically non-mutagenic, they are considered markers for genome fragility and somatic mutation rates[6]. BLM antagonizes SCE formation by dissolving double Holliday junction structures during HR, along with its partners TOPO3α, RMI1, and RMI2[7,8]. BLM also promotes regression of stalled replication forks, facilitating fork restart and preventing fork collapse and the formation of DSBs[9,10]. BS cells display higher numbers of γH2Ax foci[11], indicating frequent activation of the DNA damage response in the absence of BLM. It has also been reported that BS cells display elevated levels of loss of heterozygosity (LOH), due to exchanges between homologous chromosomes[12–14]. Besides its ability to regress replication forks and dissolve Holliday junctions, BLM has been shown to bind and unwind guanine-quadruplex (G-quadruplex, or G4) structures in vitro[15–17]. G4 structures are stable secondary DNA structures that form at guanine-rich DNA motifs[18,19] and are known barriers for replication fork progression[20].

Although SCEs can be used as a surrogate marker for collapsed forks and DSBs, their locations could until recently only be mapped cytogenetically at megabase resolution[21]. This approach does not allow investigations of the location and potential causes of fork stalling and recombination in BS. We recently described a single-cell sequencing-based technique, Strand-seq, which can be used to map SCEs at kilobase resolution, enabling novel studies of their locations and potential causes[22,23]. Strand-seq is a single-cell sequencing technique that relies on selective retention and sequencing of DNA template strands after DNA replication and cell division has occurred (Supplementary Fig. 1a). SCEs are detected as changes in orientation of DNA template strands inherited by daughter cells. By sequencing DNA template strands in single cells, changes in their directionality are identified and mapped to the genome at kilobase resolution (Supplementary Fig. 1a, b).

Here we show that SCEs in BLM-deficient cells occur frequently at sites of G4 motifs, especially those present in transcribed genes. Furthermore, we show that although LOH events appear to be more frequent in BLM-deficient cells, these events were exceedingly rare in our study. We propose that besides LOH, recombination at G4 motifs in transcribed genes is a major contributor to genome instability and cancer predisposition in BS.

## Results

### Mapping of SCEs using Strand-seq
To address the question of whether SCEs occur at random or at specific locations in the genome, we performed Strand-seq on a panel of eight different cell lines, four obtained from healthy donors (two primary fibroblast and two EBV transformed B-lymphocyte cell lines) and four cell lines from BS patients (two fibroblast and two B-cell lines) (see Supplementary Table 1). We confirmed that the BS cell lines displayed ~ 10-fold elevated SCE rates compared with wild type (WT) (Fig. 1a–d). Current Strand-seq libraries cover on average ~ 1–2% of the genome due to loss of DNA during preparation of single-cell sequencing libraries and uneven coverage further limits the resolution of SCE mapping. The median

resolution of individual SCE mapping was ~ 10 Kbp (Fig. 1e and Supplementary Fig. 1b) and > 95% of all SCE could be mapped to regions smaller than 100 Kb (Supplementary Table 1). These resolutions are several orders of magnitude higher than the megabase resolutions than can be achieved by conventional SCE mapping using cytogenetics[21].

We detected strong correlations between chromosome size and the number of SCEs on each chromosome (Fig. 1f, g), as one would expect if SCEs were randomly distributed on a global level. However, we also detected higher than expected numbers of overlapping SCE regions in multiple common fragile sites (CFSs), e.g., FRA3B (Fig. 1h) and FRA7B (Supplementary Fig. 1c), in the EBV-transformed cell lines. The absence of SCE hotspots in CFSs in primary fibroblasts (Table 1) suggest that this phenotype is intrinsic to EBV-transformed B-lymphocytes, perhaps as a result of replication stress induced by viral transformation[24]. This is consistent with previous observations that SCEs frequently occur in CFSs in cells undergoing replication stress, presumably due to replication fork stalling and collapse[25]. Strikingly, SCE frequencies within CFS hotspots are remarkably similar for the WT and BS cell lines (SCE were mapped to any given hotspot in ~ 2–9% of libraries), even though BS cells display 10-fold higher global SCE rates (Table 1). This suggests that BLM has a minor role in the processing of stalled or collapsed replication forks at CFSs.

### BS SCEs are enriched in transcribed genes
We next investigated the distribution of SCEs relative to specific genomic features of interest (FOIs). We developed a custom algorithm that compares SCE distributions with simulated random distributions in relation to a given FOI (see Methods section). For each cell line, we performed a permutation analysis to calculate the frequency of actual SCE regions overlapping with an FOI and compared it against the expected background frequency. This analysis yields relative SCE enrichments for a given FOI and allows for statistical assessment of the strength of the association.

We first turned to transcribed genes, as transcriptional activity is a known cause of genome instability and mutations through transcription–replication collisions and the formation of co-transcriptional R-loops[26,27]. BLM unwinds R-loops and the absence of BLM has been linked to genome instability at sites of R-loops[28,29]. To study a possible link between SCE locations and transcriptional status we assessed the transcriptional activity in each of our 8 cell lines using RNA-seq. Genes were divided into two categories based on the number of fragments per kilobase of processed transcript per million fragments mapped (FPKM) values: transcribed (FPKM > 1) and non-transcribed (FPKM < 1), resulting in an average of 60% (~ 23,000) of all genes classified as transcribed and 40% (~ 16,000) as non-transcribed. A significant enrichment of SCE regions overlapping with gene bodies was found in all BS cell lines, but in none of the WT cell lines (Fig. 2a). However, these enrichments were not affected by gene activity (Supplementary Fig. S2a, b). The same results were seen after subsampling SCE regions from each cell line with the lowest number of SCEs (WT1), indicating the detected SCE enrichments are not an analysis artifact caused by the higher numbers of BS SCEs (Supplementary Fig. 2c). SCEs were also significantly enriched in the gene promoter region of BS cells, independent of the transcriptional status of the associated genes (Supplementary Fig. 2d–f). We also investigated if gene expression levels affected SCE occurrence within those genes. To do this, we divided all expressed genes into four categories based on their RPKM-values, ranging from low to high expression, and assessed the number of SCEs overlapping the genes in each category. We found only weak-to-moderate correlations between gene expression levels

and SCE occurrence in all eight cell lines ($R^2$-values ranging from 0.05 to 0.64), with no differences between the WT and BS cell lines (Supplementary Fig. 2g). These results indicate that transcription by itself does not appear to have a strong role in SCE formation.

**BS SCEs are enriched at G4 motifs**. We next considered the possibility that the intragenic SCE enrichments might be caused by the presence of G4 in and around genes. BLM is known to bind and unwind G4 structures in vitro[15–17] and G4 motifs occur frequently within gene bodies and promoters[30,31]. To assess SCE

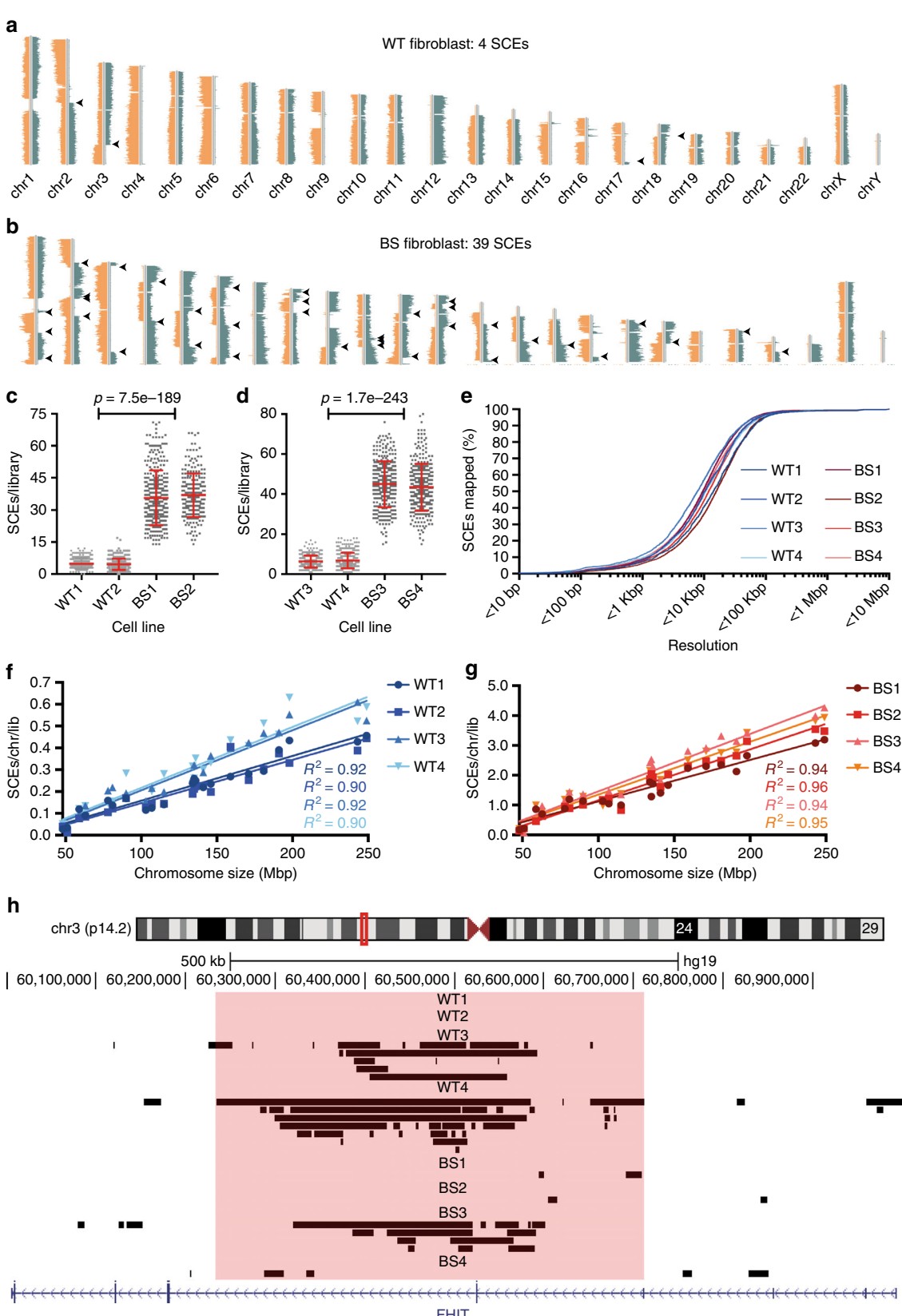

enrichments at G4 motifs, we determined the distributions of the canonical G4 motif ($G_{3+}N_{1-7}G_{3+}N_{1-7}G_{3+}N_{1-7}G_{3+}$) across the genome using a custom algorithm, and performed our SCE enrichment analysis on these regions. For this analysis, we used a stringent 10 Kb size cutoff for SCE regions included in this analysis because G4 motifs occur frequently throughout the genome (~ 8.6 Kb on average) and including larger SCE regions would result in increased noise because of the high likelihood of (permutated) SCE regions overlapping G4 motifs purely due to their size. Strikingly, we found significant, ~ 20% enrichments over expected levels for SCE regions overlapping G4 motifs in the BS cells, but no enrichments in the WT cells (Fig. 2b), indicating that G4 structures are a causal factor for SCE formation in absence of BLM. We subsequently tested if presence of G4 motifs in genes had an effect on SCE enrichments by splitting all genes into those with and those without G4 motifs. Although we detected significant SCE enrichments in BS cells for both genes with and without G4 motifs, these enrichments were stronger for genes containing at least one G4 motif (Supplementary Fig. 3a, b), indicating that the presence of G4 motifs is at least partially responsible for the SCE enrichments detected in genes in BS cells. A similar result was seen for SCE overlapping promoters with or without G4 motifs (Supplementary Fig. 3c, d). Based on these results, we decided to further investigate the link between G4 motifs, transcription, and SCE formation in BS cells.

We detected > 350,000 canonical G4 motifs in the human genome, consistent with previously reported numbers[32]. However, cells may harbor only ~ 10,000 actual G4 structures[33]. As our analysis is based on SCEs overlapping with G4 motifs, we likely overestimate overlaps with G4 structures in our permutation analysis, leading to reduced enrichment estimates. The high

prevalence of G4 motifs also means that larger SCE regions are likely to overlap with at least one G4 motif purely by chance in our permutation analysis, leading to elevated noise in our analysis and reducing relative enrichment values of the observed SCE regions. Using less stringent size cutoffs for SCE regions size than the 10 Kb cutoff used for Fig. 2b did indeed decrease the relative SCE enrichment values for the BS cells, but not the WT cells (Supplementary Fig. 4a), although BS SCE enrichments remained significant for all cutoffs used. Next, we added increasingly large flanking regions to the observed SCE regions to increase random overlaps, potentially decreasing SCE enrichment values. We did indeed observe an inverse relationship between SCE enrichments and the size of flanking regions in BS, but not WT cells lines (Supplementary Fig. 4b). This result suggests that even at our stringent 10Kbp cutoff for SCE region size, there is noise present in the permutation analysis. Taken together, we conclude that actual SCE enrichments at G4 structures are almost certainly much higher than reported in this study. As including larger SCE regions only affects SCE enrichments for BS cell lines, we also conclude that the enrichments we detect are indeed specific for BS cells.

Besides the canonical G4 motif, we also tested alternative G4 motifs for SCE enrichments. We did detect BS SCE enrichments at sites containing G4 motifs containing smaller (n1–3) and larger (n1–12) spacer regions (Supplementary Fig. 4b, c), although the enrichments were not as strong as for the canonical motif. This suggests that the canonical G4 motif is more likely to form G4 structures and induce SCE formation in vivo, or that BLM displays some specificity for G4 structures with medium-sized loops. Significant SCE enrichments were also detected at previously described "observed quadruplex regions" (Supplementary Fig. 4d), reported to constitute all regions in the genome capable of forming

**Table 1 SCE hotspots in common fragile sites, related to Fig. 1**

| Cell line | CFS | Coordinates SCE hotspot | Genes(s) | Libraries with SCE in hotspot | SCEs (n) | p-value |
|---|---|---|---|---|---|---|
| WT3 | FRA3B | chr3:60,235,449–60,601,404 | *FHIT* | 15/334 (4.5%) | 2128 | 3.2e−18 |
| | FRA7B | chr7:5,880,409–6,120,310 | *CCZ1, PMS2* | 11/334 (3.3%) | 2128 | 4.8e−13 |
| | FRA7B | chr7:6,707,575–6,967,206 | *PMS2CL, CCZ1B* | 29/334 (8.7%) | 2128 | 8.9e−50 |
| WT4 | FRA3B | chr3:60,235,449–60,601,404 | *FHIT* | 28/320 (8.8%) | 2326 | 1.5e−42 |
| | FRA5H | chr5:58,828,828–59,116,208 | *PDE4D* | 10/330 (4.0%) | 2326 | 3.8e−10 |
| | FRA7J | chr7:69,775,932–69,923,390 | *AUTS2* | 8/330 (2.4%) | 2326 | 8.2e−9 |
| | FRA16D | chr16:78,463,763–78,719,484 | *WWOX* | 8/330 (2.4%) | 2326 | 3.6e−7 |
| BS3 | FRA1A | chr1:10,628,012–10,771,196 | *PEX14, CASZ1* | 13/326 (4.0%) | 14,667 | 4.2e−9 |
| | FRA3B | chr3:60,235,449–60,601,404 | *FHIT* | 13/326 (4.0%) | 14,667 | 1.3e−4 |
| | FRA10G | chr10:52,320,213–52,640,424 | *ASAH2B, A1CF* | 14/326 (4.3%) | 14,667 | 3.2e−6 |
| | FRA16D | chr16:78,463,763–78,719,484 | *WWOX* | 13/326 (4.0%) | 14,667 | 2.8e−6 |
| BS4 | FRA7B | chr7:5,880,409–6,120,310 | *CCZ1, PMS2* | 11/334 (3.3%) | 2128 | 4.8e−13 |
| | FRA7B | chr7:6,707,575–6,967,206 | *PMS2CL, CCZ1B* | 11/306 (3.6%) | 12,724 | 5.0e−5 |
| | FRA11G | chr11:103,881,556–104,600,858 | *PDGFD* | 17/306 (5.6%) | 12,724 | 2.4e−5 |
| | FRA17q21 | chr17:43,517,064–43,840,133 | *PLEKHM1, LRRC37* | 10/306 (3.3%) | 12,724 | 6.7e−3 |

Overview of all SCE hotspots detected in human WT and BS cell lines, as well as frequency of SCE occurrence within hotspots. SCE hotspots were only detected in EBV-transformed cell lines and all hotspots occurred within known CFSs.

**Fig. 1** High-resolution mapping of SCEs and common fragile site hotspots. **a, b** Representative Strand-seq libraries generated from **a** a WT fibroblast and **b** a BS fibroblast. Mapped DNA template strand reads are plotted on directional chromosome ideograms; reads mapping to the Crick (positive) strand of the reference genome are shown in green, those mapping to the Watson (negative) strand are shown in orange. SCEs are identified as a switch in template strand state, indicated by arrowheads. **c, d** Number of SCEs detected during a single cell cycle in **c** primary fibroblasts and **d** EBV-transformed B-lymphocytes obtained from healthy donor and BS patients. Each grey point represents number of SCEs detected in a single-cell Strand-seq library, red lines indicate mean ± SD. *p*-values were calculated using ANOVA. **e** SCE mapping resolutions across all eight cell lines. Lines represent percentage of the total number of SCEs mapped at resolutions below indicated values. **f, g** Correlations between average numbers of SCEs/chromosome/library and chromosome size for **f** WT and **g** BS cells. $R^2$-values are color-matched to the cell lines. **h** Example of SCE hotspot detected within FRA3B (*FHIT*). Mapped SCE regions for each cell line were uploaded onto the UCSC Genome Browser. Black bars represent genomic locations of SCE regions; size indicates mapping resolution using the BAIT program. Red box indicates the location of the SCE hotspot as detected by Strand-seq

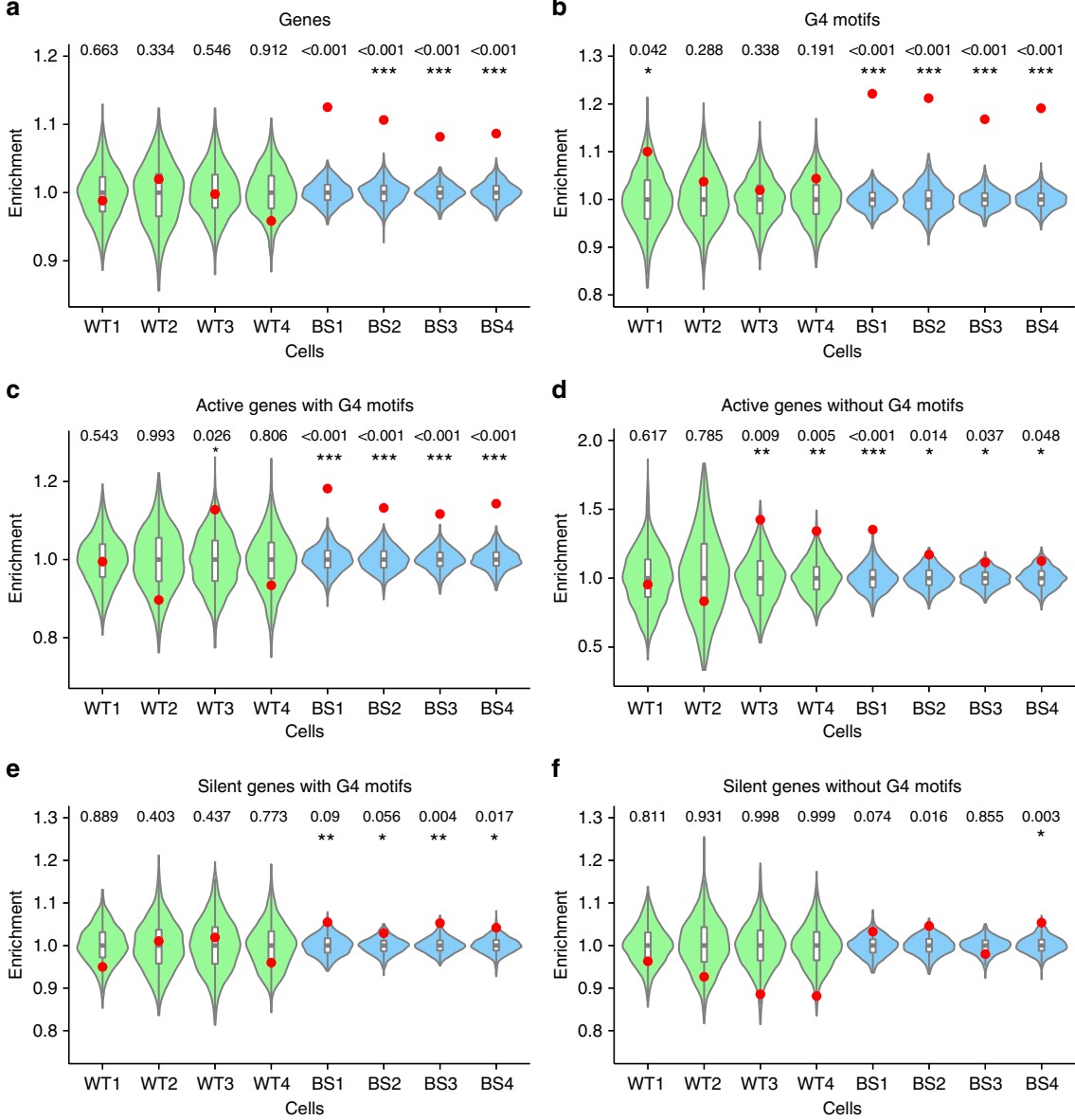

**Fig. 2** Bloom syndrome SCEs are enriched at G4 motifs in active genes. Relative SCE enrichments (red points) over random distributions (violin plots) for SCEs that overlap with one or multiple **a** genes; **b** G4 motifs ($G_{3+}N_{1-7}G_{3+}N_{1-7}G_{3+}N_{1-7}G_{3+}$); **c** active genes containing one or more G4 motifs; **d** active genes without G4 motifs; **e** silent genes containing one or more G4 motifs; and **f** silent genes without G4 motifs. All values were normalized to the median permuted value for overlap of SCEs with FOIs (out of 1,000 permutations) and relative SCE enrichments over these values were plotted on the y-axis. *p*-values indicate the fraction of permuted overlaps (out of 1,000 permutations) equal to or higher than overlap with observed SCE regions. Significant *p*-values are indicated as follows: \**p* < 0.05, \*\**p* < 0.01, \*\*\**p* < 0.001

quadruplex structures[34]. As before, SCE enrichments were not affected by SCE subsampling (Supplementary Fig. 4e). We could also exclude that enrichments were caused by nucleotide slippage or high GC content, as SCEs were specifically depleted in genomic regions with A-rich motifs ($A_{3+}N_{1-7}A_{3+}N_{1-7}A_{3+}N_{1-7}A_{3+}$) or high GC content across all eight cell lines (Supplementary Fig. 4f, g). Taken together, these results support that G4 structures are a major cause of SCE formation in BS cells.

**BS SCEs map to G4 motifs on transcribed strands**. As transcription can promote the formation of G4 structures[18] and G4s were shown to occur mainly in euchromatic regions of the genome[33], we hypothesized that BS SCEs occur at G4 motifs in transcribed genes. We therefore divided all genes into four

categories based on (1) whether genes are active or silent and (2) the presence or absence of intragenic G4 motifs, and performed a separate SCE enrichment analysis for each category. We detected the strongest BS SCE enrichments in transcribed genes containing at least one G4 motif, whereas non-transcribed genes lacking G4 motifs did not show any significant SCE enrichment patterns (Fig. 2c–f). This points to a synergistic effect of transcriptional activity and the presence of G4 motifs in genes on the enrichment of SCEs in BS cells.

Intragenic G4 motifs can occur either on the transcribed or non-transcribed strand (Fig. 3a) and it is believed that this G4 motif 'strandedness' affects how G4 structures influence gene expression[35]. To assess whether G4 strandedness affects SCE formation, we separated all intragenic G4 motifs into different categories based on strandedness and transcriptional status of the

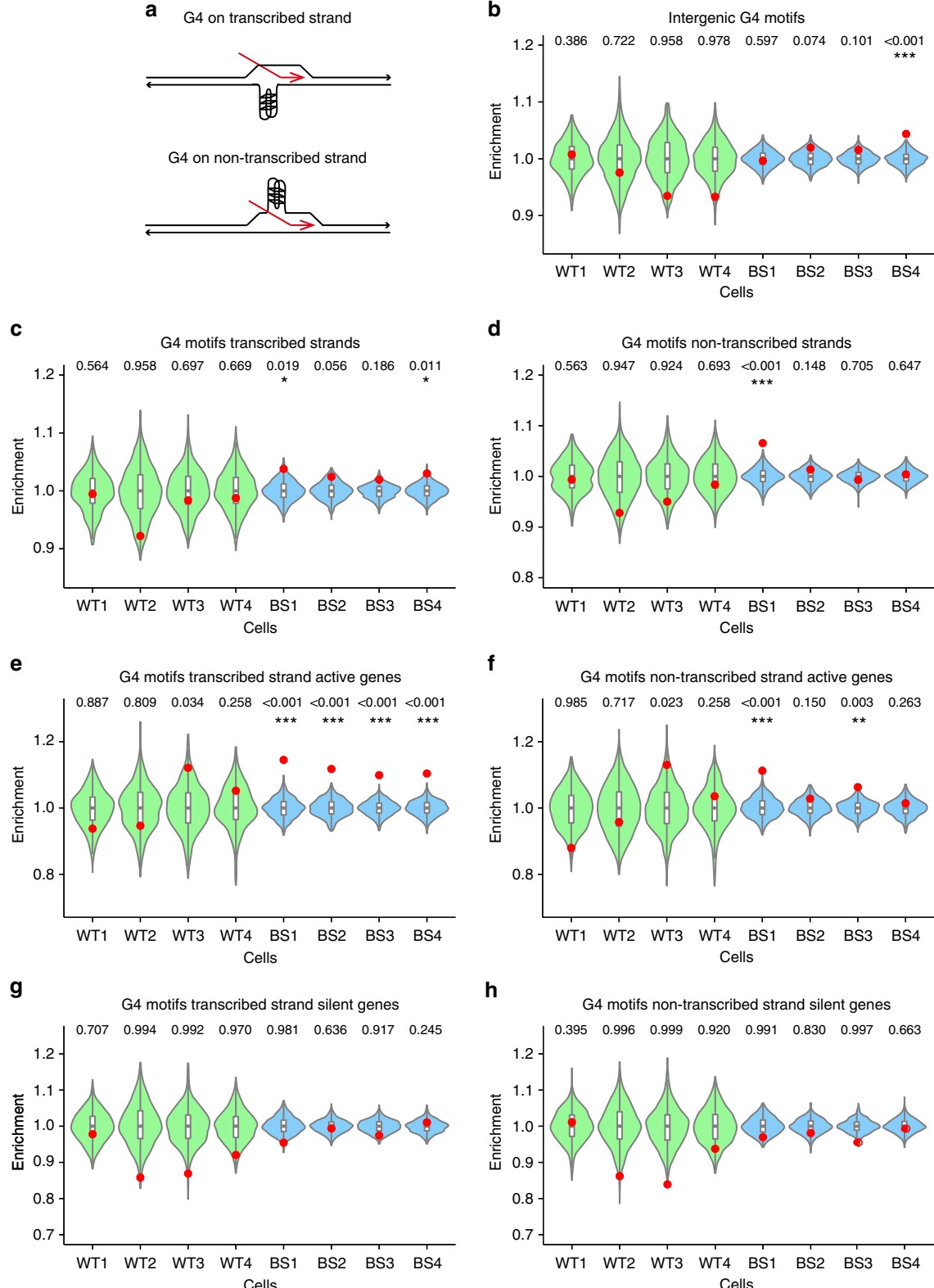

**Fig. 3** Bloom syndrome SCE enrichments occur at transcribed G4 motifs. **a** Intragenic G4 motifs can occur on either the transcribed strand (RNA shown in red), or on the non-transcribed strand; **b**–**h** relative SCE enrichments (red points) over random distributions (violin plots) for SCEs overlapping with **h** for intergenic G4 motifs, and G4 motifs occurring on **c** intragenic transcribed strands; **d** non-transcribed strands; **e** transcribed strands of active genes; **f** non-transcribed strands for active genes; **g** transcribed strands for silent genes; and **h** non-transcribed strands for silent genes. *p*-values indicate the fraction of permuted overlaps (out of 1,000 permutations) equal to or higher than overlap with observed SCE regions. Significant *p*-values are indicated as follows: \**p* < 0.05, \*\**p* < 0.01, \*\*\**p* < 0.001

gene, and performed SCE enrichment analysis for these locations. Although we found no evidence of SCE enrichments at intergenic G4 motifs (Fig. 3b), SCE are enriched at intragenic G4 motifs on both transcribed and non-transcribed strands (Fig. 3c, d).

Furthermore, BS-specific SCE enrichments were higher on transcribed than on non-transcribed strands, this effect is even strongest for G4 motifs on active transcribed genes (Fig. 3e, f). Strikingly, no SCE enrichments were detected for either

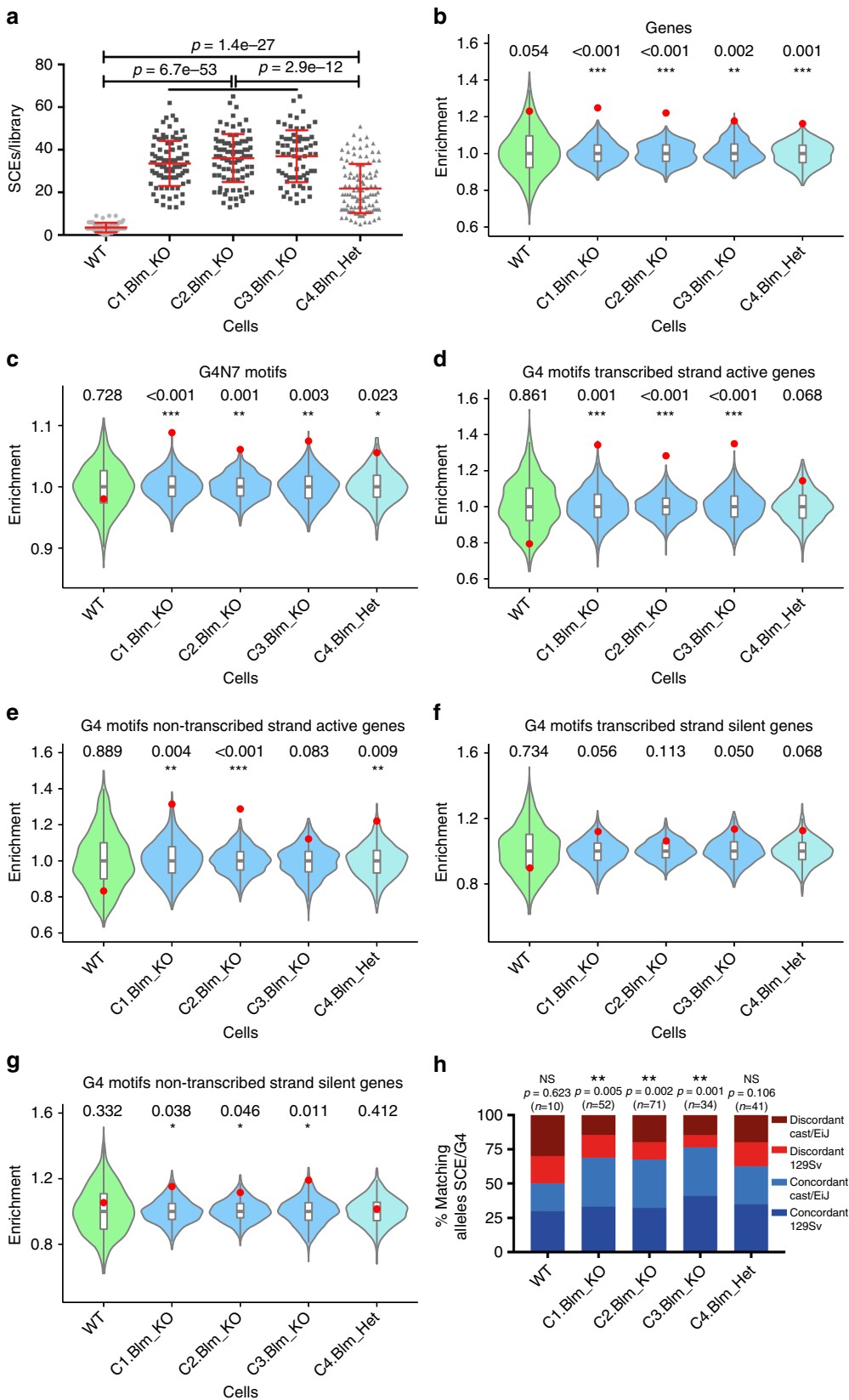

transcribed or non-transcribed strand G4 motif in silent genes (Fig. 3g, h). These results confirm the synergistic effect of transcriptional activity and the presence of a G4 motif as a trigger for SCE formation in BS cells.

**SCEs map to G4 motifs in both human and murine BLM$^{-/-}$ cells.** To confirm that the BS SCE enrichment patterns we detected in human cells are a direct result of BLM deficiency, we next generated Blm knockout cells in an F1 hybrid mouse embryonic stem (ES) cell line (129Sv-Cast/EiJ) by means of the Crispr/Cas9 technology. We used different combinations of two guide RNAs to generate loss-of-function mutants by deleting Blm exon 19, which is critical for Blm's role in both Holliday junction resolution[36] and G4 unwinding[37]. We selected three homozygous and one heterozygous clones with the desired deletions and characterized these deletions by Sanger sequencing (Supplementary Table 2), measured Blm mRNA expression levels by quantitative reverse-transcriptase PCR (qRT-PCR) (Supplementary Fig. 5a), and confirmed the elevated SCE rates by Strand-seq (Fig. 4a and Supplementary Fig. 5b, c). Interestingly, we detected intermediately high SCE rates in the Blm$^{+/-}$ cells, even though previous studies reported that cells from heterozygous family members of BS patients display normal SCE levels[2,38]. Similar to that for the human cells, SCEs in libraries made from the ES cells could be mapped at kilobase resolution (Supplementary Table 2).

Using the identified SCE regions, we performed the same analysis as described above for the human cell lines. As before, we generated RNA-seq data for each of ES cell clones to assess the effect of transcriptional activity and G4 strandedness on SCE enrichments. Although we did not detect any clear increased SCE enrichments in genes for the Blm mutant cell lines (Fig. 4b) or an effect of transcriptional activity (Supplementary Fig 5d, e), we did confirm that these cells display SCE enrichments at canonical and alternative G4 motifs (Fig. 4c and Supplementary Fig. 6a, b). We detected significant SCE enrichments at sites of intragenic G4 motifs occurring on both transcribed and non-transcribed strands in the absence of Blm (Supplementary Fig. 6c, d) and confirmed that SCE enrichments in absence of Blm are strongest at G4 motifs occurring on transcribed strands in active genes (Fig. 4d–g). As in the human cell lines, we found no SCE enrichments at sites of intergenic G4 motifs (Supplementary Fig. 6e).

The F1 hybrid ES cells we used to generate our Blm mutants contain over 20 million known heterozygous positions, including 72,660 canonical G4 motifs that only occur on one homolog (36,547 in the 129 Sv background, and 36,203 in the Cast/EiJ background). To find further evidence of a direct link between G4s and SCEs, we identified all observed SCE regions that overlap a single discordant G4 motif, and the homologs that these SCEs occurred on. We found that on average, 69% of informative SCEs in the Blm$^{-/-}$ cell lines occurred on the same homolog as the G4

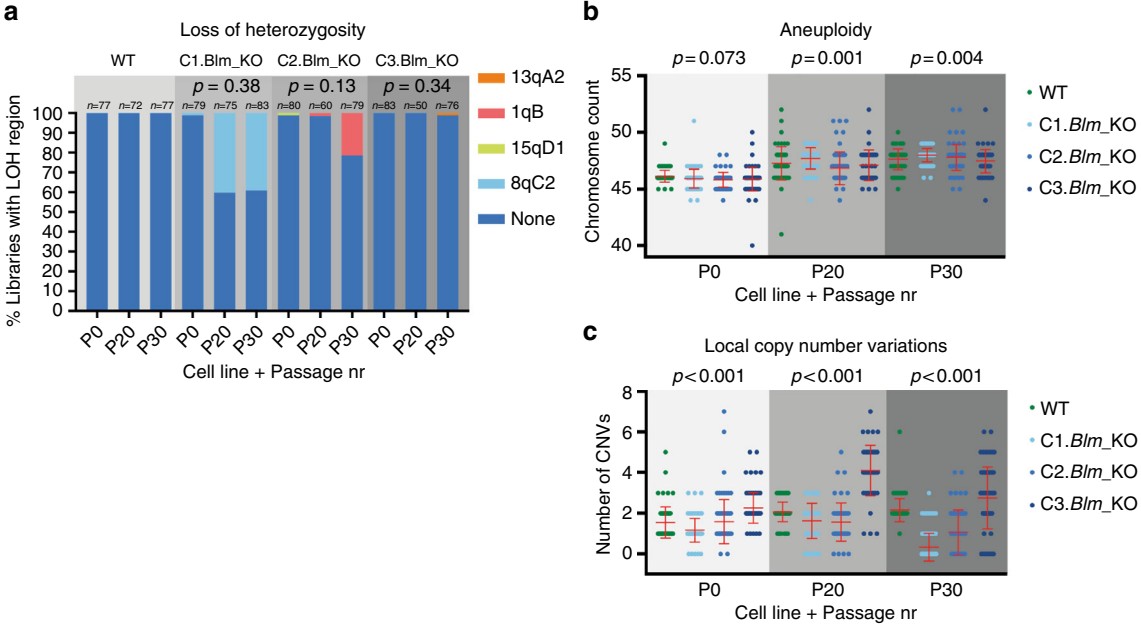

**Fig. 5** Low levels of loss of heterozygosity in Blm$^{-/-}$ mouse ES cells. Frequency of **a** unique LOH regions; **b** aneuploidy; and **c** local copy number variations detected at in single-cell whole genome sequencing libraries at different passages in WT, Blm$^{-/-}$, and Blm$^{+/-}$ mouse ES cells. The number of single-cell sequencing libraries included in the analysis is shown above each bar. p-values for LOH events were calculated using binomial distributions, for aneuploidy and CNVs by ANOVA

**Fig. 4** Confirmation of SCE enrichments at G4 motifs in Blm$^{-/-}$ mouse ES cells. **a** SCE rates detected WT, Blm$^{-/-}$, and Blm$^{+/-}$ mouse ES cells. Each grey point represents number of SCEs detected in a single-cell Strand-seq library, red lines indicate mean ± SD. p-values were calculated using t-test and ANOVA. **b–g** Relative SCE enrichments (red points) over random distributions (violin plots) for SCEs overlapping one or more (**b**) genes; **c** G4 motifs; and G4 motifs occurring on **d** transcribed strands of active genes; **e** non-transcribed strands for active genes; **f** transcribed strands for silent genes; and **g** non-transcribed strands for silent genes. p-values indicate the fraction of permuted overlaps (out of 1,000 permutations) equal to or higher than overlap with observed SCE regions. Significant p-values are indicated as follows: *p < 0.05, **p < 0.01, ***p < 0.001. **h** Frequency of observed SCE regions occurred on the same homolog as allele-specific G4 motifs. Indicated is the homolog containing G4 motif, concordant indicates SCE occurred on same homolog, discordant indicates SCE occurred on opposite homolog. Number of allelic G4 motifs included in analysis is shown above each bar. p-values were calculated using binomial distributions based on a 50% chance of SCE and G4 motif occurring on the same homolog

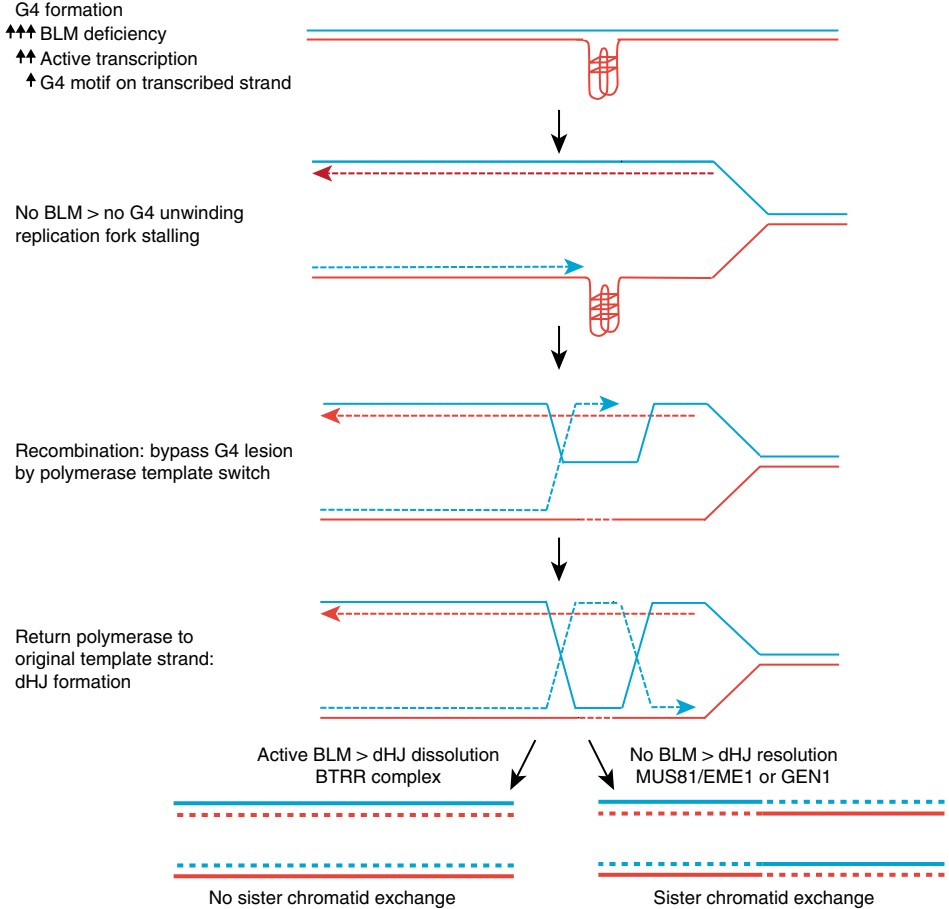

**Fig. 6** BLM helicase suppresses recombination at G4 structures. Model for the role of BLM in suppressing recombination at sites of G4 structures. G4 structures are more likely to form or persist in the absence of unwinding by BLM. They can form at G4 motifs throughout the genome, but formation is promoted by transcription, especially if the G4 motif is present on the transcribed strand. In BLM proficient cells, BLM unwinds the G4 structure before the genomic region is replicated, ensuring smooth DNA replication. In the absence of BLM, G4 structures are not unwound, preventing replication fork progression, and leading to replication fork stalling. Stalled forks require homologous recombination (HR)-mediated repair, leading to formation of a double Holliday junction (dHJ). In the absence of BLM, dHJs cannot be dissolved by the BLM-TOPO3α-RMI1-RMI2 (BTRR) complex and must be resolved by MUS81-EME1 or GEN1, leading to frequent formation of sister chromatid exchanges event, and potentially loss of heterozygosity, and other types of mutations

motifs, which is significantly different ($p < 0.01$) from the expected 50% if there was no causal relationship between G4 motifs and SCEs (Fig. 4h). No significant deviation from the expected 50/50 ratio was detected in the WT or the $Blm^{+/-}$ cell lines. Combined, these results confirm that SCEs mainly form at G4 structures in absence of Blm and especially at those G4s present in the transcribed strands of active genes.

**LOH is not significantly increased in $Blm^{-/-}$ cells.** As SCEs are exchanges of genetic material between identical sister chromatids, they normally do not result in any mutations. However, if an exchange event occurs between homologs instead of sister chromatids, this can lead to LOH[39]. It has previously been shown that BLM deficient cells display elevated levels of LOH[12–14]. However, these results were obtained using systems that rely on selection of cells that underwent LOH at a specific locus. Using our F1 ES cell lines, we could detect and track LOH events throughout the entire genome based on single-nucleotide polymorphisms between the parental mouse strains. To do this, we kept the WT and $Blm$ mutant ES cells in continuous culture for 30 passages (~ 75 cell divisions), which would result in $3.8 \times 10^{22}$ offspring cells for each parental cell, compared to an estimated $1.2 \times 10^{10}$ cells in an adult mouse body. We performed single-cell whole-genome sequencing

(scWGS) at different timepoints (passages 0, 20, and 30), and identified chromosomal regions that underwent LOH (see Methods). We also identified chromosomal and local copy number variations (CNVs) to confirm that LOH regions are not caused by deletions, and to determine if the $Blm^{-/-}$ cells display aberrant levels of CNVs.

We did not detect a single LOH region in the WT cells at any of the three time points, and only four unique LOH regions in the three $Blm^{-/-}$ clones (Fig. 5a). Two of these four regions were detected in a single library at a single time point, while the two others were detected at multiple time points and their frequency increased over time. However, these more frequent LOH regions occurred on chromosomes 1 and 8, both of which display increasing levels of trisomy in all four cell lines (Supplementary Fig. 7). This suggests that trisomy led to clonal expansion within the cell populations, and the detected LOH regions had no effect on cellular proliferation. Although these results do point towards elevated LOH in $Blm^{-/-}$ cells, the differences are not significant and suggest that LOH is an uncommon occurrence, even in the absence of Blm.

BLM has been linked to chromosome segregation[40] and BLM-deficient cells display a higher frequency of micronuclei[41], both of which can result in aneuploidy. When we assessed the WT and $Blm^{-/-}$ cells for instances of local and chromosomal CNVs, we

found that although there are significant differences between the individual cell lines, no trend can be seen indicating that $Blm^{-/-}$ cells contain more or fewer such events (Fig. 5b, c).

## Discussion

Elevated SCE rates are a hallmark feature in cells from BS patients[2,3], but the exact mechanism behind this phenotype is not fully understood. A major obstacle to unravelling the cause of BS SCEs was that SCEs cannot be accurately mapped using standard cytogenetic detection methods. For this study, we used Strand-seq for SCE detection, as this technique does allow for high-resolution mapping. Even though the technique is limited by loss of DNA during preparation of single-cell sequencing libraries, leading to low coverage within individual libraries (~ 1–2% genome coverage), we show here that SCEs in both normal and BS cells could be mapped at kilobase resolutions, allowing for robust analysis on SCE locations and thus their causes.

We show that SCEs frequently occur at sites of G4 structures in both BLM deficient human and murine cells. While there does not appear to be a direct effect of transcriptional activity on SCE enrichment patterns, strong SCE enrichments in BS cells were observed in transcribed genes containing one or more G4 motifs, especially when the G4 motif was present on the transcribed DNA strand. The observation that SCEs were enriched on homolog-specific G4 motifs in the $Blm$ mutant ES cells provides further evidence that, in the absence of BLM, G4 structures can directly trigger SCE formation.

Studies of G4 structures have been hampered by their high stability, making them resistant against several nucleases[42] and difficult to analyze using standard PCR conditions[34]. The use of Strand-seq bypasses these issues, because SCE regions are identified as the region between sequencing reads. As such, any SCE overlap with G4 motifs requires that the G4 motif lies within the identified SCE region and therefore does not have to be covered by sequencing reads itself.

Previous studies have shown that BLM is required for unwinding G4 structures during telomere replication[43], and that it has a role in regulating expression of genes containing G4 motifs[44,45]. Our study is the first to directly implicate G4 structures in the increased recombination and genome instability in cells that lack BLM. These results are consistent with proposed models of BLM unwinding G4 structures during DNA replication[37,46] and with previous reports that BLM binds and unwinds G4 structures in vitro[15,16]. Our results show that BLM is required to unwind G4 structures throughout the genome. G4 structures are known to pose barriers for DNA replication[20] and previous studies have shown that specialized helicases such as Dog-1, Pif1 and FANCJ are required to prevent instability at G-rich genomic DNA in *Caenorhabditis elegans*[47], yeast[48], and man[49]. The fact that such other helicases cannot compensate for loss of BLM suggests that these helicases do not have redundant functions, but are either specific for subsets of G4 structures, or that they cooperate to unwind G4 structures, as was proposed for BLM and FANCJ[50,51]. Consistent with this, BLM deficient cells display elevated levels of G4 structures at telomeres[43], and it seems logical that this holds true throughout the genome. We propose that failure to unwind G4 structures in BLM-deficient cells leads to stalled replication forks, which trigger recombination and genome instability (Fig. 6).

SCEs in cells lacking BLM were found to frequently occur in transcribed genes, supporting that such sites are subject to higher mutation rates[27]. Elevated intragenic mutations rates are likely to contribute to the strong cancer predisposition associated with BS. This also helps explain a unique feature of BS, which predisposes patients to a wide range of cancers instead of towards specific

types of tumors[1]. Combined with elevated LOH levels, the proposed chromosome fragility is likely to play a role in the strong cancer predisposition associated with the syndrome.

## Methods

**Cell cultures**. The following cell lines were obtained from the Corriell Cell Repository: GM07492 and GM07545 (primary fibroblasts, normal), GM02085 and GM03402 (primary fibroblasts, BS), GM12891 and GM12892 (EBV-transformed B-lymphocytes, normal), and GM16375 and GM17361 (EBV-transformed B-lymphocytes, BS). The WT hybrid mouse ES cell line F121.6 (129Sv-Cast/EiJ) was a kind gift from Joost Gribnau (Erasmus University, Rotterdam, The Netherlands).

Fibroblasts were cultured in Dulbecco's modified Eagle's medium (DMEM) (Life Technologies) supplemented with 10% v/v fetal bovine serum (FBS) (Sigma Aldrich) and 1% v/v penicillin–streptomycin (Life Technologies), B-lymphocytes in RPMI1640 (Life Technologies) supplemented with 15% v/v FBS and 1% v/v penicillin–streptomycin.

ES cells were cultured on mitotically arrested mouse embryonic fibroblast cells in DMEM (Life Technologies), supplemented with 15% v/v FBS (Bodinco BV), 1% v/v penicillin–streptomycin, 1% v/v non-essential amino acids (Life Technologies), 50 μM 2-mercaptoethanol (ThermoFisher Scientific), and 1,000 U ml$^{-1}$ leukemia inhibitor factor (Merck). All cells were cultured at 37 °C in 5% CO$_2$. For Strand-seq, BrdU (Invitrogen) was added to exponentially growing cultures at 40 μM final concentration. Timing of BrdU pulse was 12 h for ES cells, 18 h for fibroblast cell lines, and 24 h for B-lymphocyte cell lines.

**Generation of Blm mutant ES cell lines**. Blm mutants were generated using CRISPR/Cas9 genome editing. sgRNAs were designed to cleave the *Blm* gene at sites flanking exon 19 and cloned into PX459 plasmid[52]. Combinations of two plasmids (30 μg each) were transfected into F121.6 cells by means of electroporation (Biorad Genepulser XL). Cells were incubated for 24 h before puromycin (1 μg/ml) was added to cell culture medium. After 48 h of selection, resistant colonies were left to grow, picked and expanded. Screening for *Blm* mutant clones was performed by allele-specific PCR of genomic region containing putative deletion.

**qRT-PCR analysis**. Exponentially growing cells were collected and RNA was isolated using the Nucleospin RNA kit (Macherey Nagel). Reverse transcription was performed using Superscript II Reverse Transcriptase (Invitrogen) with random hexamers (Invitrogen). Quantitative PCR was performed using SYBR Green I Master (Roche) on the LightCycler480 (Roche).

**Strand-seq and scWGS library preparation**. For Strand-seq and WGS, exponentially growing cells were collected after BrdU pulse (for Strand-seq) or without any treatment (WGS), and resuspended in nucleo isolation buffer (100 mM Tris-HCl pH 7.4, 150 mM NaCl, 1 mM CaCl$_2$, 0.5 mM MgCl$_2$, 0.1% NP-40, and 2% bovine serum albumin) supplemented with 10 μg ml$^{-1}$ Hoechst 33,258 (Life Technologies) and propidium iodide (Sigma Aldrich). Single nuclei were sorted into 5 μl Pro-Freeze-CDM NAO freeze medium (Lonza) + 7.5% dimethyl sulfoxide, in 96-well skirted PCR plates (4Titude), based on low propidium iodide and low Hoechst fluorescence using a MoFlo Astrios cell sorter (Beckman Coulter) or a FACSJazz cell sorter (BD Biosciences). DNA from single cells was processed for Strand-seq[23] or WGS[53]. For each experiment, 96 libraries were pooled and 250–450 bp-sized fragments were isolated and purified. DNA quality and concentrations were assessed using the High Sensitivity dsDNA kit (Agilent) on the Agilent 2100 Bio-Analyzer and on the Qubit 2.0 Fluorometer (Life Technologies).

**RNA-seq library preparation**. Exponentially growing cells were harvested and RNA was isolated using the Nucleospin RNA kit (Macherey Nagel). RNA-sequencing libraries were prepared using the NEBNext Ultra RNA Library Prep kit for Illumina (NEB) combined with the NEBNext rRNA Depletion kit (NEB). Complementary DNA quality and concentrations were assessed using the High Sensitivity dsDNA kit (Agilent) on the Agilent 2100 Bio-Analyzer and on the Qubit 2.0 Fluorometer (Life Technologies).

**Illumina sequencing**. Clusters were generated on the cBot (HiSeq2500) and single-end 50 bp reads (Strand-seq and RNA-seq) or paired-end 150 bp reads (scWGS) were generated were generated using the HiSeq2500 sequencing platform (Illumina).

**Bioinformatics**

*Genome alignment*. Indexed bam files were aligned to human (GRCh37) or mouse genomes (GRCm38) using Bowtie2[54] for Strand-seq and scWGS libraries, and STAR aligner[55] for RNA-seq libraries.

*Sister chromatid exchange detection*. SCE were identified and mapped with the BAIT software package[56], using standard settings. As BAIT also detects stable chromosomal rearrangements, events that occurred at the exact same locations in

> 5% of cells from one cell line were excluded from the analysis. SCEs were assigned to homologs by splitting.bam files into separate files for each genetic background based on reads covering informative polymorphisms and using BAIT to identify on which homologs SCEs occurred.

*Detection and analysis of SCE hotspots.* BAIT-generated.bed files containing the locations of all mapped SCEs were uploaded to the USCS genome browser and hotspots were identified as regions containing multiple overlapping SCEs. *p*-values were assigned to putative SCE hotspots using a custom R-script based on capture–recapture statistics. Briefly, the genome was divided into bins of the same size as the putative hotspot and the chance of findings the observed number of SCEs in one bin was calculated based on the total number of SCEs detected in the cell line.

*Enrichment analysis.* A custom Perl script was used for the permutation model. For each of 1,000 permutations, we generated a random number *n* and shifted all SCEs downstream by *n* bases on the same chromosome. To prevent small-scale local shifts, we required *n* to be a random number between 2 and 50 Mbp. If the resulted coordinate exceeded chromosome size we subtracted the size of chromosome, so that the SCE is mapped to beginning part of the chromosome, as if the chromosome was circular. We also excluded all annotated assembly gaps before our analysis, to prevent permuted SCE mapping to one of the gap regions. We then determined the number of SCEs overlapping with a feature of interest in each permutation, as well as the original SCE regions. All values were normalized to the median permutated value, in order to determine relative SCE enrichments over expected, randomized distributions and to allow for comparison of the different cell lines. Significance was determined based on how many permutations showed the same or exceeding (enrichment) or the same or receding (depletion) overlap with a given genomic feature compared to overlap between the original SCEs and the same feature. Any experimental overlap that lies outside of the 95% confidence interval found in the permutations has a *p*-value below 0.05 and was deemed significant. Experimental overlaps lying outside of the permuted range were given a *p*-value below 0.001, as there was a <0.1% (1/1,000) chance of such an overlap occurring by chance. Enrichment analyses for G4 motifs were performed using a 10 Kb SCE region size cutoff, enrichment analysis for genes and promoter regions used a 100 Kb size cutoff, unless specified otherwise. Genome and gene annotations were obtained from Ensembl release 75 (GRCh37 assembly, http://www.ensembl.org). Gene bodies were defined as regions between transcription start sites and transcription end sites, gene promoters as 1 Kbp regions upstream of transcription start sites. Putative G4 motifs were predicted using custom Perl script by matching genome sequence against following patterns: $G_{3+}N_xG_{3+}N_xG_{3+}N_xG_{3+}$, where *x* could be the ranges of 1–3, 1–7, or 1–12 bp.

*RNA-seq analysis.* Mapped reads were aligned and quantified using STAR aligner[55]. FPKM values were calculated for all genes and based on these genes were assigned active (FPKM > 1) or silent (FPKM < 1) status.

*Aneuploidy and CNV detection.* Aligned libraries were analyzed as previously described using AneuFinder R package[57] using the following settings: low-quality alignments (mapping quality score (MAPQ) < 10) and duplicate reads were excluded and read counts in 2 Mb variable-width bins were determined with a 10-state Hidden Markov Model with copy-number states: zero-inflation, null-, mono-, di-, tri-, tetra-, penta-, hexa-, septa-, and octasomy.

*LOH detection.* Reads were aligned to either 129 Sv or Cast/EiJ genetic background based on covered single-nucleotide polymorphisms (SNPs). Reads lacking informative SNPs were discarded. Reads (129 Sv) were assigned a positive (Crick) orientation, Cast/EiJ reads a negative (Watson) orientation. The resulting.bam files were analyzed using BAIT and LOH events were detected as switches from mixed background to pure 129 Sv or Cast/EiJ background in the absence of deletions (as detected using AneuFinder).

**Data availability**. The Strand-seq, scWGS, and RNA-seq data reported in this paper have been submitted to the Arrayexpress database under accession E-MTAB-5976. SCE enrichment analysis software is available through GitHub (https://github.com/Vityay/GenomePermute).

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

## Acknowledgements

We thank Dirk Hockemeyer, Marcel van Vugt, and Peter Stirling for critical reading of this manuscript, Inge Kazemier and Karina Hoekstra-Wakker for technical assistance, and Ester Falconer, Mark Hills, and all members of the Lansdorp laboratories in Vancouver and Groningen for discussions and feedback. Financial support was provided by an Advanced Grant from the European Research Council to P.M.L.

## Author contributions

N.v.W. and P.M.L. conceived and designed the study. N.v.W. and S.M. created and characterized Blm mutant cell lines. N.v.W. and N.H. performed Strand-seq, scWGS, and RNA-seq experiments. D.C.S.J. supervised next-generation sequencing efforts. N.v.W. and V.G. analysed sequencing data. N.v.W. wrote the manuscript with assistance from S.M., P.M.L. and all authors. P.M.L. supervised the project.

## Additional information

**Competing interests:** The authors declare no competing financial interests.

