## [Peer Review File · Nature Communications]

Reviewers' Comments:

Reviewer #1:

Remarks to the Author:

Lansdorp due 10-26

This manuscript applies the ingenious Strand-seq method developed by Lansdorp and collaborators (Sanders et al. Nat Protocol 2017; Ref 23) to mapping SCE's in BLM-deficient vs. normal human cells. A panel of 8 cell lines was analyzed, including fibroblast lines from 2 normal and 2 BLM-deficient individuals, and EBV-transformed B lymphocyte lines from 2 normal and 2 BLM-deficient individuals. Statistically strong correlations are documented between chromosome size and the number of SCEs per chromosome; and between SCEs and the presence of a G4 motif, especially at transcribed genes. These are significant and timely results, which should be of interest to the readers of Nature Communications. However, there are some questions and issues of data presentation that need to be addressed before the manuscript is ready for publication.

Strand-seq provides sequence information on the strand that served as the template for replication in cells cultured for one generation in BrdU, which creates DNA duplexes that are "hemi-labeled". Single nuclei are sorted, DNA nicked with micrococcal nuclease, bar-coded linkers added, the BrdU-labeled strand destroyed by treatment with Hoechst 33258 and UV, and DNA amplified, pooled and sequenced. The Nat Protocol paper presents a thoughtful discussion of strengths and weaknesses of the method. Strand-seq can define haplotypes and provide a close-up of anomalies that are difficult to detect by other sequencing methods, such as inversions, translocations and — especially important for the manuscript at hand — SCEs. However, the many steps are accompanied by losses of material that currently limit coverage to 1-5%. The losses and limited coverage characteristic of Strand-seq raise the concern that regions that form G4 structures may be over- or under-represented among the final products. G4 structures are resistant to many kinds of nucleases, and G4 may be untouched in the MNase step, or if cleaved it may not unfold for linker addition. Similarly, G4 structures are notoriously difficult to PCR, and this can lead to under-representation. Nuclease resistance is a particular concern because libraries of nascent strands that identified G4 at replication origins subsequently came under criticism because the lambda exo used for library construction does not digest G4. If possible, these points should be addressed experimentally, for example by showing that recovery of G4 with different loop lengths reflects abundance in the genome. If not, these caveats should be raised in the Discussion.

Questions and comments:

It would be useful for the introduction or first section of results to provide more information on Strand-seq and its strengths and weaknesses. There could even be a diagram as part of Supp Fig 1.

G4 motifs are enriched at promoters. The results for frequencies (or rates) of SCEs in promoter regions and elsewhere should be presented not only as raw data, but also normalized to abundance of G4 motifs in the region being examined, as otherwise significance could be under- or over-estimated.

SCEs form at CFSs independent of BLM status. Is the correlation between SCE formation and G4 increased if SCEs at CFSs are removed from the database?

The 10 kb cutoff for proximity of a G4 motif to an SCE used in Fig. 2b seems reasonable to me, but the authors should provide a plausible biological basis for this cutoff in the text in the context of that figure. It would also be very informative to learn how results change if the cutoff is dropped.

The use of the terms template, transcribed and coding strand are confusing (Figures 3, 4 and elsewhere). Strand-seq data are presumably all from the template strand for the final round of replication. I would strongly recommend consistently using transcribed and non-transcribed to distinguish strands of active genes. This helps the reader from confusing transcription and replication and also avoids giving the impression that genes must encode something to be included in the dataset.

p. 5: Does the absence of BLM-dependence of SCE formation at CFSs really “contradict established models” for BLM counteracting fork stalling, as stated? This seems like overstatement, especially without explicitly distinguishing between those established models and the model presented in Fig. 6.

Do the transcribed genes participating in SCEs correlate with genes shown to depend upon BLM for transcription, where there was also a correlation with G4 (Ref. 42)?

The LOH results (Fig. 5 and text) are not strong enough for publication, since there are not enough examples to evaluate their statistical significance.

Table 1 would be clearer if highlighting identified all BS results, rather than alternate lines.

Comparisons between Table 1 and Supp Table 1 would be facilitated if data columns present the same parameters in the same order.

Figures and Figure Legends

For all figures: place parentheses indicating which panel is described before, not after, the description of that panel.

For all figures containing violin plots — Figures 2, 3 and 4, and several supplementary figures:

1. State how p values were calculated.
2. Increase font size of labels indicating p values so they are legible at standard magnification. This would especially strengthen Figure 4, where dots do not emerge from the violin shading.
3. Add asterisks to identify results that are significant.
4. The violin plots as currently colored obscure the center of the diagram and the red dots in some cases. Perhaps make coloring lighter or remove coloring and color background instead?

Figure 1c-d: are these SCE rates per generation? Time constant should be stated.

Figure 1b: the color described as green is gray in the pdf version of the figure.

Figure 1e: less than half the x-axis contains useful distinctions – perhaps focus on central region and eliminate the rest?

Figure 1f: this single example of mapping SCEs in a fragile site is interesting. It would be useful to include additional examples in a supplementary figure, especially as it shows that this example is representative.

Figure 1, legend: sense/antisense are confusing descriptions when applied to the physical genome, as they are open to alternative interpretations determined by the transcriptional direction of the gene.

Figure 2b: the proximity necessary to call a G4 motif at/near an SCE should be restated in legend. Define median permuted value in figure legend.

Figure 2a: are the data shown for all RefSeq genes?

Figure 4: It would be useful to add a diagram at the top, like that on Figure 3.

Figure 5: legend should identify x axis labels (passage number?). Numbers of events should be identified.

Are any of these observations statistically significant? If not, they should be eliminated as they detract from the strong central claim of the ms.

Figure 6: several comments:

1. This model would be more powerful if uncoupled from transcription, and it applies to both transcribed and non-transcribed regions.
2. Don't describe G4 as a lesion but a structure – since it does no harm if resolved by BLM.
3. Indicate sites at which dHJ cleavage is predicted to occur to generate an SCE.
4. Do events need to occur on the leading strand to create an SCE? If so, explain why; if not, diagram or explain for the lagging strand.

Supplementary Figures

The results in Figure S1b, c, which show that there is a strong correlation between chromosome size and frequency of SCEs, are central to the argument and could be included in the main body of figures, for example as part of Figure 1.

p.5: "We detected strong correlations between chromosome size and the number of SCEs on each chromosome (Supp Fig. 1c-d)" — there is no Supp Fig. 1d. It seems that the intention was to refer to Supp Figs 1b and c, which show R2 values for plots of SCEs vs chromosome size. The Supp Fig. Legends need correction too, as the legends refer to these panels as Supp Figs. 1d-e, and no legends are not provided for Supp Figs. 1b, 1c.

Reviewer #2:

Remarks to the Author:

The authors have employed Strand-seq, a single-cell sequencing method developed by Dr. Lansdorp's group, to study the role of the Bloom syndrome helicase (BLM) in sister chromatid exchange (SCEs). BLM is best known as a component of the BTR dissolvase, which acts to suppress recombination. The elevation in SCEs is thought to play a key role in the cellular phenotype of BS cells. So, new insights into how BLM might affect these events, especially at the whole genome level, are welcome.

The experimental design involved use of eight cell lines: two derived from primary fibroblasts, two derived from EBV-transformed B-lymphocytes, two BS fibroblast lines, and two BS B-cell lines. In this way, the authors could use Strand-seq to measure relative changes in SCE frequency and location as a function of BLM expression.

A major goal of the paper was to investigate SCEs in BS cells at a resolution that has hitherto been impossible to achieve. The Strand-seq method relies upon selective degradation of the nascent DNA after a single cell division, followed by sequencing of the parental DNA, and it provides improved resolution for the determination of SCE location. This is one major contribution of the current work.

The manuscript does a nice job of presenting a series of interesting observations derived from the Strand-seq data sets that provide new insights into the role of BLM and G4 structures in the maintenance of genomic stability. The experimental design, statistical analyses, and interpretation of the results are all sound and presented in a clear manner.

A key component of the study was the use of customized Perl scripts that allowed comparison of enrichment analysis for different genomic features of interest (FOIs) over different SCE region size

cutoffs, with about half of the SCEs mapped to regions of 10Kbp and >95% of SCEs mapping to regions of 100Kbp, which is still an improvement over the resolution of previous assays. Honing in on genomic FOIs affecting SCE enrichment (e.g. genes, transcribed genes, gene promoters, etc.), the authors identified G quadruplex motifs as a causal element in the elevated levels of SCEs observed in BLM-deficient cells. The relationship between RecQ helicases, such as BLM, and G quadruplexes is well established through numerous biochemical and cell-based studies. Loss of RecQ helicase function is known to impact transcriptional profiles, telomere maintenance, and G quadruplex replication. With that said, the specific localization of SCEs to G4 sites in BS cells is another major contribution of the study and likely to be of interest to many scientists in the fields of DNA replication and repair, as well as researchers in other fields of chemistry and biology.

Comments:

1. Page 5: The authors detected overlap between elevated SCE formation and common fragile sites in EBV-transformed cells, but report no difference between WT and BS cell lines. They go on to state that this contradicts “established models of BLM actively counteracting SCE formation at stalled replication forks.” This statement is rather broad. The sentence should probably end “at common fragile sites.”. The authors are clearly aware that CFSs represent a subset of endogenous barriers to replication. Indeed, G4 motifs are sources of fork slowing/stalling. Perhaps I am misinterpreting what the authors are trying to state here, but it seems to me that saying BLM does not participate in the suppression of SCEs at “stalled forks” in general is not justified.

2. Some of the conclusions drawn from the current work are informed by and expand upon the G4-ChIP study reported by the Balasubramanian group (ref 31). This previous report identified some 10K G4 structures in immortalized cells, indicative of the fact that chromatin suppresses the formation of the remaining >300K G4 motifs. That same study found much fewer actual G4 structures (~1K) in “normal” keratinocytes. It is possible that BS cells have a larger number of G4 structures actually formed compared to WT cells and that this contributes to the elevated SCE frequency. Indeed, previous studies have shown that loss of BLM induces epigenetic instability at G4 sites, which could lead to more facile G4 folding in regions of the genome that would otherwise suppress such an event. In a way, the authors have examined this idea by comparing canonical and non-canonical G4 motifs (page 8) but perhaps the authors could comment a bit more directly on the possibility that higher levels of folded G4 structures (derived from the canonical G4 motif) could impact SCE frequency.

3. The recent G4-ChIP study (ref 31) reported G4 structure enrichment in “highly” transcribed genes. The comparison of BS SCE enrichment in either transcribed or non-transcribed genes is informative. It would be interesting to know if the SCE overlap in WT and BS cells was dependent upon the relative expression level, as opposed to binning the genes as either active or silent. Such an analysis may not be feasible with the current data sets. Alternatively, gene set analysis of sites with elevated SCEs could reveal enrichment of specific subsets of genes where G4 sites induce genomic instability in BS cells (e.g. cancer related genes such as c-MYC).

4. Is there a way the authors can breakdown the fraction of SCEs at different genomic regions associated with G4 potential (e.g. telomeres, ribosomal DNA, cancer related genes)? Based on the results shown in Figure 1, it would seem that some portion of the SCEs observed in BLM-deficient cells are located near the telomeres. Given the previously established role for BLM in telomere maintenance, some commentary on this subject is warranted.

Minor comments:

1. Page 6 – artifact is misspelled near the bottom of the page.

2. Page 7, second paragraph – add references for the line noting >350,000 G4 motifs in the human genome.

3. Page 7, bottom of the page – change “. . . to increased random overlaps,” to read “to increase random overlaps”.

4. Page 8: Change "...to form G4 structures and induced SCE formation" to read "...to form G4 structures and induce SCE formation"
5. Page 14: supporting is misspelled
6. Page 14: The authors should provide a reference to support the statement that transcribed genes are "subject to higher mutation rates."
7. Page 14, end of first sentence: The authors should include a reference to work from Dr. Julian Sale's group implicating BLM/WRN and FANCD1 in replication of G4 motifs (Sarkies et al. 2012 Nucleic Acids Res. 40, p. 1485-98).
8. The authors might consider publishing the Perl scripts as supporting information.

Response to reviewers

Reviewer #1 (Remarks to the Author):

This manuscript applies the ingenious Strand-seq method developed by Lansdorp and collaborators (Sanders et al. Nat Protocol 2017; Ref 23) to mapping SCE's in BLM-deficient vs. normal human cells. A panel of 8 cell lines was analyzed, including fibroblast lines from 2 normal and 2 BLM-deficient individuals, and EBV-transformed B lymphocyte lines from 2 normal and 2 BLM-deficient individuals. Statistically strong correlations are documented between chromosome size and the number of SCEs per chromosome; and between SCEs and the presence of a G4 motif, especially at transcribed genes. These are significant and timely results, which should be of interest to the readers of Nature Communications. However, there are some questions and issues of data presentation that need to be addressed before the manuscript is ready for publication.

We thank the reviewer for these encouraging comments and the overall positive evaluation of our paper. We very much appreciate her/his detailed comments which have allowed us to greatly improve the paper.

Strand-seq provides sequence information on the strand that served as the template for replication in cells cultured for one generation in BrdU, which creates DNA duplexes that are “hemi-labeled”. Single nuclei are sorted, DNA nicked with micrococcal nuclease, bar-coded linkers added, the BrdU-labeled strand destroyed by treatment with Hoechst 33258 and UV, and DNA amplified, pooled and sequenced. The Nat Protocol paper presents a thoughtful discussion of strengths and weaknesses of the method. Strand-seq can define haplotypes and provide a close-up of anomalies that are difficult to detect by other sequencing methods, such as inversions, translocations and — especially important for the manuscript at hand — SCEs. However, the many steps are accompanied by losses of material that currently limit coverage to 1-5%. The losses and limited coverage characteristic of Strand-seq raise the concern that regions that form G4 structures may be over- or under-represented among the final products. G4 structures are resistant to many kinds of nucleases, and G4 may be untouched in the MNase step, or if cleaved it may not unfold for linker addition. Similarly, G4 structures are notoriously difficult to PCR, and this can lead to under-representation. Nuclease resistance is a particular concern because libraries of nascent strands that identified G4 at replication origins subsequently came under criticism because the lambda exo used for library construction does not digest G4. If possible, these points should be addressed experimentally, for example by showing that recovery of G4 with different loop lengths reflects abundance in the genome. If not, these caveats should be raised in the Discussion.

The reviewer raises some excellent points regarding the difficulty to capture G4-DNA in sequencing libraries. However, SCE regions are identified as regions flanked by DNA template strand reads of opposite orientation (see Supplementary Figure 1). Our enrichment analysis relies on the presence or absence of G4 motifs within this same region of the reference genome. As such, our analysis does not require the G4 motifs to be covered in by Strand-seq libraries. These points are now included in the discussion section, and we have included additional explanation on the principles of Strand-seq in Supplementary Figure 1 (see also our reply to the next comment).

Questions and comments:

It would be useful for the introduction or first section of results to provide more information on Strand-seq and its strengths and weaknesses. There could even be a diagram as part of Supp Fig 1.

We have included an additional diagram in Supplementary Figure 1 explaining the principles of Strand-seq, and have expanded the figure legend to include more detail. We also included a brief comment on low coverage and how this affects SCE mapping resolution in the discussion section of the manuscript.

G4 motifs are enriched at promoters. The results for frequencies (or rates) of SCEs in promoter regions and elsewhere should be presented not only as raw data, but also normalized to abundance of G4 motifs in the region being examined, as otherwise significance could be under- or over-estimated.

We have now expanded our analysis by including SCE enrichments for all genes and promoters separated by the presence of G4 motifs (Supplementary Figure 3).

SCEs form at CFSs independent of BLM status. Is the correlation between SCE formation and G4 increased if SCEs at CFSs are removed from the database?

We performed this analysis and did not see notable differences between SCE enrichments if SCEs occurring in CFS hotspots were removed (see Figure 1 below). This is perhaps not surprising, as SCEs in CFS hotspots only represent a minority of the total number of SCEs in each cell line (especially in the BS cell lines).

The 10kb cutoff for proximity of a G4 motif to an SCE used in Fig. 2b seems reasonable to me, but the authors should provide a plausible biological basis for this cutoff in the text in the context of that figure. It would also be very informative to learn how results change if the cutoff is dropped.

The 10Kb cutoff was chosen based on the average distance between G4 motifs in the genome (~8.6Kb). As we show in Supplementary Figure 4a-b, including larger SCE regions in our analysis has a negative effect on SCE enrichments due to increased noise in the analysis. We have included an additional analysis in this graph where no size cutoff was used and explain our reasoning behind using the 10Kb cutoff in the manuscript.

The use of the terms template, transcribed and coding strand are confusing (Figures 3, 4 and elsewhere). Strand-seq data are presumably all from the template strand for the final round of replication. I would strongly recommend consistently using transcribed and non-transcribed to distinguish strands of active genes. This helps the reader from confusing transcription and replication and also avoids giving the impression that genes must encode something to be included in the dataset.

We have made these changes: template strands are now referred to as ‘transcribed’, and coding strands as ‘non-transcribed’, genes are now classified as ‘active’ or ‘silent’ instead of ‘transcribed’ or ‘non-transcribed’.

p. 5: Does the absence of BLM-dependence of SCE formation at CFSs really “contradict established models” for BLM counteracting fork stalling, as stated? This seems like overstatement, especially without explicitly distinguishing between those established models and the model presented in Fig. 6.

We agree that this statement does not properly reflect the message we tried to confer, and have changed it accordingly (see also comment by reviewer 2).

Do the transcribed genes participating in SCEs correlate with genes shown to depend upon BLM for transcription, where there was also a correlation with G4 (Ref. 42)?

The study in question identified 1153 genes with significant differential expression between healthy and BS fibroblasts, but no distinction was made between changes in expression of active genes and genes being turned on or off in absence of BLM. We have now included a new analysis showing that relative gene expression levels have no significant effect on SCE rates (Supplementary Fig 2g), and that the presence or absence of G4 motifs in genes and promoters does have an impact on SCE enrichments (Supplementary Fig 3). In a way, this matches the observation that differentially expressed genes frequently contain one or more G4 motifs.

The LOH results (Fig. 5 and text) are not strong enough for publication, since there are not enough examples to evaluate their statistical significance.

We agree that the frequency of LOH events in WT and BS cells is too low to establish a statistical difference. However, contrary to (our) expectation we found a very low frequency of LOH in both WT and *Blm*^{-/-} cells. We have changed the text to emphasize the low frequency of LOH in *Blm*^{-/-} cells.

Table 1 would be clearer if highlighting identified all BS results, rather than alternate lines. Comparisons between Table 1 and Supp Table 1 would be facilitated if data columns present the same parameters in the same order.

These tables have been rearranged to increase clarity.

Figures and Figure Legends

For all figures: place parentheses indicating which panel is described before, not after, the description of that panel.

This has been amended.

For all figures containing violin plots — Figures 2, 3 and 4, and several supplementary figures:

1. State how p values were calculated.
2. Increase font size of labels indicating p values so they are legible at standard magnification. This would especially strengthen Figure 4, where dots do not emerge from the violin shading.
3. Add asterisks to identify results that are significant.
4. The violin plots as currently colored obscure the center of the diagram and the red dots in some cases. Perhaps make coloring lighter or remove coloring and color background instead?

All violin plots have been redone and should now be clearer. Explanations of p-values have been added to each figure legend, font sizes for p-values have been increased as much as possible within the available space, and significant p-values are now marked with an asterisk, with explaining in the figure legends. Coloring of violin plots has been changed for increased clarity on screen, and in both full-color and black-and-white prints.

Figure 1c-d: are these SCE rates per generation? Time constant should be stated.

These numbers represent SCE that occurred during a single cell cycle, this is now stated in the figure legend.

Figure 1b: the color described as green is gray in the pdf version of the figure.

The color is green on the pdf file we generated, perhaps some coloring was lost during processing of our submission. We will be sure to keep an eye on this in the proofs.

Figure 1e: less than half the x-axis contains useful distinctions – perhaps focus on central region and eliminate the rest?

We agree that visible differences between the cell lines only occur in the middle section of the x-axis. We chose to display the entire axis to highlight 1) the size distribution of SCE regions, 2) that differences between the cell lines are minor, and 3) that there are no consistent differences between the WT and BS cell lines.

Figure 1f: this single example of mapping SCEs in a fragile site is interesting. It would be useful to include additional examples in a supplementary figure, especially as it shows that this example is representative.

We have included another example of a fragile site hotspot in Supplementary Figure 1.

Figure 1, legend: sense/antisense are confusing descriptions when applied to the physical genome, as they are open to alternative interpretations determined by the transcriptional direction of the gene.

We have removed these terms and now only refer to DNA strands as Watson (negative) and Crick (positive).

Figure 2b: the proximity necessary to call a G4 motif at/near an SCE should be restated in legend. Define median permuted value in figure legend.

G4 motifs need to overlap with identified SCE regions for at least 1 nucleotide to be included in our analysis, this is now stated in the text.

Figure 2a: are the data shown for all RefSeq genes?

For this analysis we used all annotated genes from the Ensembl database, this is now stated in the methods section.

Figure 4: It would be useful to add a diagram at the top, like that on Figure 3.

We understand that this would be useful, but Figure 4 is already the size of a full page in print. Readers can refer back to Figure 3 if needed.

Figure 5: legend should identify x axis labels (passage number?). Numbers of events should be identified. These changes have been made.

Are any of these observations statistically significant? If not, they should be eliminated as they detract from the strong central claim of the ms.

Statistical analysis has been performed on LOH frequency in WT vs $Blm^{-/-}$ cells, the results are now included in the figure. No significant differences were detected, which contradicts previously published results referenced in the text. We have made changes to the text to emphasize this. We also redid our analysis of aneuploidy and CNV frequency and included statistical analysis in Fig. 5.

Figure 6: several comments:

1. This model would be more powerful if uncoupled from transcription, and it applies to both transcribed and non-transcribed regions.
2. Don't describe G4 as a lesion but a structure – since it does no harm if resolved by BLM.
3. Indicate sites at which dHJ cleavage is predicted to occur to generate an SCE.
4. Do events need to occur on the leading strand to create an SCE? If so, explain why; if not, diagram or explain for the lagging strand.

We have made several changes to this figure and the accompanying text to more clearly explain the proposed model. We have reduced the emphasis on transcription, and removed references to leading and lagging strand replication.

Supplementary Figures

The results in Figure S1b, c, which show that there is a strong correlation between chromosome size and frequency of SCEs, are central to the argument and could be included in the main body of figures, for example as part of Figure 1.

These results have been moved to Figure 1.

p.5: “We detected strong correlations between chromosome size and the number of SCEs on each chromosome (Supp Fig. 1c-d)” — there is no Supp Fig. 1d. It seems that the intention was to refer to Supp Figs 1b and c, which show R2 values for plots of SCEs vs chromosome size. The Supp Fig. Legends need correction too, as the legends refer to these panels as Supp Figs. 1d-e, and no legends are not provided for Supp Figs. 1b, 1c.

This has been corrected.

Reviewer #2 (Remarks to the Author):

The authors have employed Strand-seq, a single-cell sequencing method developed by Dr. Lansdorp's group, to study the role of the Bloom syndrome helicase (BLM) in sister chromatid exchange (SCEs). BLM is best known as a component of the BTR dissolvase, which acts to suppress recombination. The elevation in SCEs is thought to play a key role in the cellular phenotype of BS cells. So, new insights into how BLM might affect these events, especially at the whole genome level, are welcome.

The experimental design involved use of eight cell lines: two derived from primary fibroblasts, two derived from EBV-transformed B-lymphocytes, two BS fibroblast lines, and two BS B-cell lines. In this way, the authors could use Strand-seq to measure relative changes in SCE frequency and location as a function of BLM expression.

A major goal of the paper was to investigate SCEs in BS cells at a resolution that has hitherto been impossible to achieve. The Strand-seq method relies upon selective degradation of the nascent DNA after a single cell division, followed by sequencing of the parental DNA, and it provides improved resolution for the determination of SCE location. This is one major contribution of the current work.

The manuscript does a nice job of presenting a series of interesting observations derived from the Strand-seq data sets that provide new insights into the role of BLM and G4 structures in the maintenance of genomic stability. The experimental design, statistical analyses, and interpretation of the results are all sound and presented in a clear manner.

A key component of the study was the use of customized Perl scripts that allowed comparison of enrichment analysis for different genomic features of interest (FOIs) over different SCE region size cutoffs, with about half of the SCEs mapped to regions of 10Kbp and >95% of SCEs mapping to regions of 100Kbp, which is still an improvement over the resolution of previous assays.

Honing in on genomic FOIs affecting SCE enrichment (e.g. genes, transcribed genes, gene promoters, etc.), the authors identified G quadruplex motifs as a causal element in the elevated levels of SCEs observed in BLM-deficient cells. The relationship between RecQ helicases, such as BLM, and G quadruplexes is well established through numerous biochemical and cell-based studies. Loss of RecQ helicase function is known to impact transcriptional profiles, telomere maintenance, and G quadruplex replication. With that said, the specific localization of SCEs to G4 sites in BS cells is another major contribution of the study and likely to be of interest to many scientists in the fields of DNA replication and repair, as well as researchers in other fields of chemistry and biology.

We thank the reviewer for these encouraging comments and the overall positive evaluation of our paper. We very much appreciate her/his detailed comments which have allowed us to greatly improve the paper.

Comments:

1. Page 5: The authors detected overlap between elevated SCE formation and common fragile sites in EBV-transformed cells, but report no difference between WT and BS cell lines. They go on to state that this contradicts "established models of BLM actively counteracting SCE formation at stalled replication forks." This statement is rather broad. The sentence should probably end "at common fragile sites.". The authors are clearly aware that CFSs represent a subset of endogenous barriers to replication. Indeed, G4 motifs are sources of fork slowing/stalling. Perhaps I am mis-interpreting what the authors are trying to state here, but it seems to me that saying BLM does not participate in the suppression of SCEs at "stalled forks" in general is not justified.

We agree that this statement does not properly reflect the message we tried to convey, and have changed it accordingly (see also comment by reviewer 1).

2. Some of the conclusions drawn from the current work are informed by and expand upon the G4-ChIP study reported by the Balasubramanian group (ref 31). This previous report identified some 10K G4 structures in immortalized cells, indicative of the fact that chromatin suppresses the formation of the

remaining >300K G4 motifs. That same study found much fewer actual G4 structures (~1K) in “normal” keratinocytes. It is possible that BS cells have a larger number of G4 structures actually formed compared to WT cells and that this contributes to the elevated SCE frequency. Indeed, previous studies have shown that loss of BLM induces epigenetic instability at G4 sites, which could lead to more facile G4 folding in regions of the genome that would otherwise suppress such an event. In a way, the authors have examined this idea by comparing canonical and non-canonical G4 motifs (page 8) but perhaps the authors could comment a bit more directly on the possibility that higher levels of folded G4 structures (derived from the canonical G4 motif) could impact SCE frequency.

The reviewer raises an excellent point, and we have included some discussion on this topic.

3. The recent G4-ChIP study (ref 31) reported G4 structure enrichment in “highly” transcribed genes. The comparison of BS SCE enrichment in either transcribed or non-transcribed genes is informative. It would be interesting to know if the SCE overlap in WT and BS cells was dependent upon the relative expression level, as opposed to binning the genes as either active or silent. Such an analysis may not be feasible with the current data sets. Alternatively, gene set analysis of sites with elevated SCEs could reveal enrichment of specific subsets of genes where G4 sites induce genomic instability in BS cells (e.g. cancer related genes such as c-MYC).

We have included an extra figure panel showing that there is no correlation between relative gene expression levels and the number of SCEs overlapping these genes (Supplementary Figure 2g).

4. Is there a way the authors can breakdown the fraction of SCEs at different genomic regions associated with G4 potential (e.g. telomeres, ribosomal DNA, cancer related genes)? Based on the results shown in Figure 1, it would seem that some portion of the SCEs observed in BLM-deficient cells are located near the telomeres. Given the previously established role for BLM in telomere maintenance, some commentary on this subject is warranted.

BLM is indeed known to play a role in telomere maintenance and ALT, and it has previously been reported that SCEs frequently occur in (sub)telomeres. Unfortunately, we cannot detect SCEs at (sub)telomeres, since their highly repetitive nature precludes accurate mapping of sequencing reads (and therefore SCEs) to those locations. Unfortunately, rDNA loci cannot be reliably mapped to the human genome using Strand-seq and without their exact locations we also cannot perform SCE enrichment analyses for these features. We did perform SCE enrichment analyses on genes containing or lacking G4 motifs separately, and included the results in Supplementary Figure 3.

Minor comments:

1. Page 6 – artifact is misspelled near the bottom of the page.

This has been corrected.

2. Page 7, second paragraph – add references for the line noting >350,000 G4 motifs in the human genome. This number was taken from our own analysis, but we now explain that our numbers are consistent with those previously reported.

3. Page 7, bottom of the page – change “.. to increased random overlaps,” to read “to increase random overlaps”.

This has been corrected.

4. Page 8: Change “..to form G4 structures and induced SCE formation” to read “...to form G4 structures and induce SCE formation”

This has been corrected.

5. Page 14: supporting is misspelled

This has been corrected.

6. Page 14: The authors should provide a reference to support the statement that transcribed genes are “subject to higher mutation rates.”

A reference for this statement has been added.

7. Page 14, end of first sentence: The authors should include a reference to work from Dr. Julian Sale's group implicating BLM/WRN and FANCD1 in replication of G4 motifs (Sarkies et al. 2012 Nucleic Acids Res. 40, p. 1485-98).

This reference has been added.

8. The authors might consider publishing the Perl scripts as supporting information.

These scripts have been uploaded to GitHub, a link is provided in the manuscript.

Figure 1. Comparative SCE enrichments for G4 motifs using SCE regions (a) before and (b) after filtering for SCEs overlapping CFS hotspots. P-values indicate the fraction of permuted overlaps (out of 1,000 permutations) equal to or higher than overlap with observed SCE regions. Significant p-values are indicated as follows: *: $p < 0.05$, **: $p < 0.01$, ***: $p < 0.001$.

Reviewers' Comments:

Reviewer #1:

Remarks to the Author:

The authors have responded exceedingly well to almost all the comments from both reviewers, but I'm still somewhat concerned about the question of the effect of BLM on LOH. It looks like the previous reports need to be revisited looking carefully for G4 motifs in the regions near the reported. In any event, the reader needs to be made aware that they must look closely at previous results and make up their own mind about this question.

So, I'd encourage the authors to consider the following changes:

p.3: Change "It has also been shown that BS cells display elevated levels of loss of heterozygosity (LOH), due to exchanges between homologous chromosomes 12-14."

To : "It has also been reported..."

p.14: Change the header "Low levels of loss of heterozygosity in Blm-/- cells" (which could even be an equivocal way of saying LOH is reduced)

To: LOH is not significantly increased in BLM-/- cells.

Minor:

p. 14: Change oncommon to uncommon

Reviewer #2:

Remarks to the Author:

The authors have done an excellent job of responding to each of my comments from the previous review. I think this is an outstanding contribution and have no additional major comments.

Minor comments:

Page 14, line 286: replace "oncommon" with "uncommon".

Page 15, line 316: "therefore" is misspelled.

Response to reviewers

Reviewer #1 (Remarks to the Author):

The authors have responded exceedingly well to almost all the comments from both reviewers, but I'm still somewhat concerned about the question of the effect of BLM on LOH. It looks like the previous reports need to be revisited looking carefully for G4 motifs in the regions near the reported. In any event, the reader needs to be made aware that they must look closely at previous results and make up their own mind about this question.

So, I'd encourage the authors to consider the following changes:

p.3: Change "It has also been shown that BS cells display elevated levels of loss of heterozygosity (LOH), due to exchanges between homologous chromosomes 12-14."

To : "It has also been reported..."

We have made this change.

p.14: Change the header "Low levels of loss of heterozygosity in Blm^{-/-} cells" (which could even be an equivocal way of saying LOH is reduced)

To: LOH is not significantly increased in BLM^{-/-} cells.

We have made this change.

Minor:

p. 14: Change uncommon to common

This has been corrected.

Reviewer #2 (Remarks to the Author):

The authors have done an excellent job of responding to each of my comments from the previous review. I think this is an outstanding contribution and have no additional major comments.

Minor comments:

Page 14, line 286: replace "oncommon" with "uncommon".

This has been corrected.

Page 15, line 316: "therefore" is misspelled.

This has been corrected.